# Retrieval Augmented Diffusion Model for Structure-informed Antibody Design and Optimization

**Zichen Wang**[1][*]     **Yaokun Ji**[1][*]     **Jianing Tian**[2]     **Shuangjia Zheng**[1][†]
[1] Global Institute of Future technology, Shanghai Jiao Tong University;
[2] School of Software & Microelectronics, Peking University

`zichen_w@sjtu.edu.cn, shuangjia.zheng@sjtu.edu.cn`

## Abstract

Antibodies are essential proteins responsible for immune responses in organisms, capable of specifically recognizing antigen molecules of pathogens. Recent advances in generative models have significantly enhanced rational antibody design. However, existing methods mainly create antibodies from scratch without template constraints, leading to model optimization challenges and unnatural sequences. To address these issues, we propose a retrieval-augmented diffusion framework, termed RADAb, for efficient antibody design. Our method leverages a set of structural homologous motifs that align with query structural constraints to guide the generative model in inversely optimizing antibodies according to desired design criteria. Specifically, we introduce a structure-informed retrieval mechanism that integrates these exemplar motifs with the input backbone through a novel dual-branch denoising module, utilizing both structural and evolutionary information. Additionally, we develop a conditional diffusion model that iteratively refines the optimization process by incorporating both global context and local evolutionary conditions. Our approach is agnostic to the choice of generative models. Empirical experiments demonstrate that our method achieves state-of-the-art performance in multiple antibody inverse folding and optimization tasks, offering a new perspective on biomolecular generative models.

## 1 Introduction

Antibodies, essential Y-shaped proteins in the immune system, are pivotal for recognizing and neutralizing specific pathogens known as antigens. This specificity primarily arises from the Complementarity Determining Regions (CDRs), which are crucial for binding affinity to antigens (Jones et al., 1986; Ewert et al., 2004; Xu & Davis, 2000; Akbar et al., 2021). The design of effective CDRs is therefore central to developing potent therapeutic antibodies, a dominant class of protein therapeutics. However, the development of these antibodies typically relies on labor-intensive experimental methods such as animal immunization or screening extensive antibody libraries, often failing to produce antibodies that target therapeutically relevant epitopes effectively. Thus, the ability to generate new antibodies with pre-defined biochemical properties in silico carries the promise of speeding up the drug design process.

Computational efforts in antibody design have traditionally involved grafting residues onto existing structures (Sormanni et al., 2015), sampling alternative native CDR loops to enhance affinities(Aguilar Rangel et al., 2022), and using tools like Rosetta for sequence design improvements in interacting regions (Adolf-Bryfogle et al., 2018). Many recent studies have focused on applying deep generative models to design antibodies (Luo et al., 2022; Martinkus et al., 2024; Zhu et al., 2024). They take advantage of geometric learning and generative models to capture the higher-

---

[*]Equal Contribution

[†]Corresponding author

order interactions among residues directly from the data. These innovations provide more efficient methods to search sequence and structure spaces.

Albeit powerful, current generative models struggle to design antibodies that adhere to structural constraints and exhibit desired biological properties. This challenge primarily arises from a lack of diversity in the available training data. Predominantly, research efforts have relied on the SAbDab database (Dunbar et al., 2014), which comprises fewer than ten thousand antigen-antibody complex structures. The limited scope of this dataset restricts the models' ability to capture comprehensive high-order interaction information between antigen-antibody residues, thereby increasing the risk of overfitting. Moreover, most existing methodologies attempt to design antibody sequences *de novo*, without the benefit of template-based guidance. This approach inherently demands a greater volume of data and extensive training or fine-tuning on specific datasets to achieve efficacy in practical applications.

In this work, we draw inspiration from template-based and fragment-based antibody design to develop a model that fully utilizes protein structural database, effective motif retrieval, and semi-parametric generative neural networks. Our goals are to: (a) leverage template-aware local and global protein geometric information to enhance model generative capability, (b) integrate motif evolutionary signals to prevent overfitting, and (c) necessitate minimal training or fine-tuning for effective use in real-world applications.

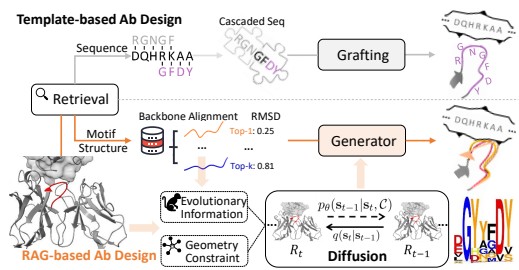

Figure 1: Illustration of the retrieval-augmented framework.

To this end, we introduce the **R**etrieval-**A**ugmented **D**iffusion **A**nti**b**ody design model (RADAb), a novel semi-parametric antibody design framework. To fully exploit the protein structure space, we first compiled a database of CDR-like fragments from the non-redundant Protein Data Bank (PDB) (Berman et al., 2000). These CDR-like fragments are linear functional motifs structurally compatible with an antibody CDR loop, found in any protein geometry within the PDB. For a given antibody to be improved, we perform a structural retrieval to obtain motifs with structures similar to the desired CDR framework. As protein sequences capable of folding into similar structures often share homology and consensus, we hypothesize that these retrieved motifs, enriched with evolutionary information, can enhance the model's generalization.

Unlike traditional rational design methods that optimize by grafting a single CDR fragment, we propose to use a set of structural homologous CDR-like motifs together with the desired backbone for iterative sequence optimization (Figure 1). Our major contributions follow: (1) We propose a **first-of-its-kind** retrieval-augmented generative framework for rational antibody design. It uses a set of functional CDR-like fragments that satisfy the desired backbone structures and properties to guide generation toward satisfying all the required properties. (2) **A novel retrieval mechanism** is introduced for integrating these exemplar motifs with the input backbone through a novel dual-branch denoising module, utilizing both structural and evolutionary information. Additionally, we present a coupled conditional diffusion module that iteratively refines the evolution process by incorporating global and local conditions. This allows the model to incorporate more functional information than traditional antibody inverse folding models. (3) Empirical experiments demonstrate that our method improves the state-of-the-art methods in multiple antibody inverse folding tasks, e.g., an 8.08% AAR gains in long CDRH3 inverse folding task and an average of 7 cal/mol absolute $\Delta\Delta G$ improvements in functionality optimization task, offering a fresh perspective on biomolecular generative models.

## 2 RELATED WORK

**Antibody Design** Computational antibody design primarily follows two paths: conventional energy function optimization methods and machine learning approaches. Early antibody design methods were often limited to sequence similarity and energy function optimization (Lapidoth et al., 2015; Adolf-Bryfogle et al., 2018). Recent success of machine learning approaches mainly falls into two directions: antibody sequence design and antigen-specific antibody sequence-structure co-design.

The methods used for antibody sequence design mainly include language-based models (Ruffolo et al., 2021; Olsen et al., 2022; Wang et al., 2023a) and inverse folding models (Dreyer et al., 2023; Høie et al., 2024). The other line focuses on antibody sequence-structure co-design mainly taking antibody-antigen complex as a graph, then using graph networks to extract features and predict the coordinates and residue type of antibody CDR (Jin et al., 2021; Kong et al., 2022; 2023; Lin et al., 2024; Luo et al., 2022; Zhu et al., 2024; Martinkus et al., 2024). While these works are undoubtedly powerful, they often generate antibodies from scratch without incorporating explicit structure constraints, which can introduce challenges in designing functional antibodies (Zhou et al., 2024). Instead, our method leverages the power of templates from a structure-informed perspective.

**Diffusion generative models** Diffusion models (Sohl-Dickstein et al., 2015; Song et al., 2020; Ho et al., 2020) are a class of generative models that have achieved impressive progress on a lot of generation tasks. Denoising diffusion probabilistic models (DDPMs) are a branch of diffusion models, which contain two Markov processes. The forward process perturbs the data into pure noise, and then learns to generate data by reversing the forward Markov process. Because of the diffusion model's flexibility and controllability, numerous works are focusing on employing retrieval-augmented methods to complement the diffusion framework for text-to-image generation (Sheynin et al., 2023), image generation (Blattmann et al., 2022), human motion generation (Zhang et al., 2023a) and small molecule generation (Huang et al., 2024).

**Retrieval augmented generative models** Retrieval augmented generation technique was first proposed in the field of natural language processing to enhance the language models by introducing an additional database (Lewis et al., 2020; Guu et al., 2020), prompting the language models to generate more realistic and diverse results. Subsequently, retrieval augmented generation (RAG) has conducted diverse explorations in large fields, including natural language processing (Zhang et al., 2023b; Gao et al., 2023; Xu et al., 2024; Yoran et al., 2024; Caffagni et al., 2024) and computer vision (Long et al., 2022; Blattmann et al., 2022; Hu et al., 2023; Rao et al., 2023).

Recently, several studies have been proposed using retrieval techniques to enhance molecular generation. RetMol (Wang et al., 2023b) generates new molecules based on existing small molecules by retrieving a set of exemplar moleculars. IRDiff (Huang et al., 2024) enhances protein-specific molecular generation by using protein pockets to retrieve moleculars which interct with the pocket. Although there has been some retrieval-based work in the field of protein design and discovery (Zhou & Grigoryan, 2015; Aguilar Rangel et al., 2022), to the best of our knowledge, this is the first-of-its-kind retrieval-based generative framework for antibody design.

## 3 PRELIMINARIES AND NOTATIONS

### 3.1 NOTATIONS

Antibody consists of two heavy chains and two light chains. Each chain's tip has a complementary site that specifically binds to a unique epitope on the antigen. This site includes six complementary-determining regions (CDRs): CDR-H1, CDR-H2, and CDR-H3 on the heavy chain, and CDR-L1, CDR-L2, and CDR-L3 on the light chain (Presta, 1992; Al-Lazikani et al., 1997).

Our work represents each single protein residue in terms of the residue type $s_i \in \{ACDEFGHIKLMNPQRSTVWY\}$, the coordinate $x_i \in \mathbb{R}^3$, and the orientation $\mathbf{O}_i \in SO(3)$, where $i = 1, ..., N$ and N is the number of residues in the complex. Concretely, assuming the CDR sequence to be generated includes $m$ amino acids and starts from position $a$, it can be denoted as $R = \{s_j \mid j \in \{a+1, ..., a+m\}\}$. Let $M$ be the length of the antibody, the antibody framework is defined as $C_{ab} = \{(s_i, x_j, \mathbf{O}_j) \mid i \in \{1, ..., M\} \backslash \{a+1, ..., a+m\}, j \in \{1, ..., M\}\}$. The antibody framework sequence is defined as $S_{ab} = \{s_i \mid i \in \{1, ..., M\} \backslash \{a+1, ..., a+m\}\}$. The antigen is defined as $C_{ag} = \{(s_i, x_i, \mathbf{O}_i) \mid i \in \{M+1, ..., N\}\}$. The retrieved CDR-like fragments are defined as $\mathbb{A} = \{A_i \mid i \in \{1, ..., k\}\}$. The goal of our framework is to extract the antibody CDR structure from the antibody framework complex $C_{ab}$, then input it into the retrieval module to retrieve $\mathbb{A}$, and ultimately predict the distribution of $R$ through $C_{ag}$, $C_{ab}$ and $\mathbb{A}$.

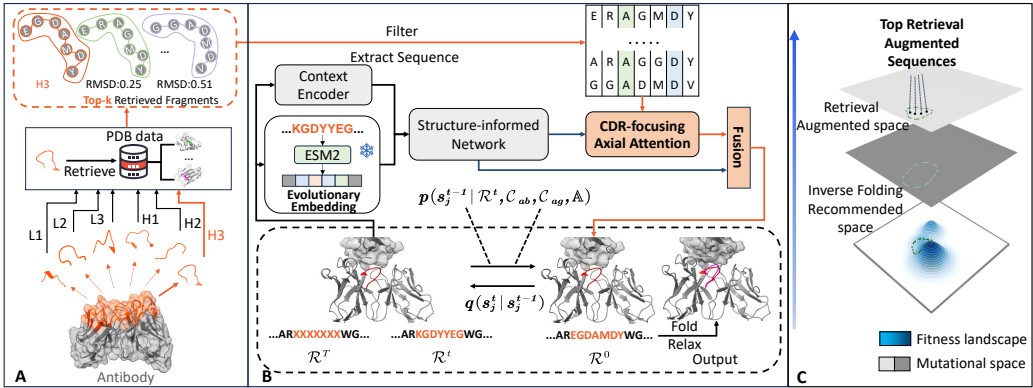

Figure 2: The overall architecture of the proposed RADAb framework. (A) Structural retrieval process, the CDR backbone is input into MASTER and the output is a set of ranked CDR-like fragments. (B) Diffusion process and denoising network which takes antibody-antigen context and retrieved evolutionary information as conditions. The structure is fixed during diffusion process. (C) Our method restricts the antibody to a small region through fixed structural constraints and retrieval-augmented constraints (functional constraints) to achieve higher fitness.

## 3.2 DIFFUSION MODEL FOR ANTIBODY DESIGN

Due to diffusion models' excellent performance and controllability, there are now many diffusion-based works that have achieved notable results (Luo et al., 2022; Villegas-Morcillo et al., 2023; Kulytė et al., 2024). To be specific, they are denoising probabilistic diffusion models that transform the amino acid type $s$, the backbone $C\alpha$ atom coordinates $x$, and the amino acid orientation $\mathbf{O}$ during the diffusion process. We focus on sequence, of which the forward process perturbs the data in the following ways (Hoogeboom et al., 2021):

$$q\left(s_j^t \mid s_j^{t-1}\right) = \text{Multinomial}\left(\left(1 - \beta^t\right) \cdot \text{onehot}\left(s_j^{t-1}\right) + \beta^t \cdot \frac{1}{20} \cdot \mathbf{1}\right) \quad (1)$$

where $\beta^t$ is the noise schedule for the diffusion process, as $t$ approaches T, $\beta^t$ will approach 1, and the probability distribution will become closer to pure noise. $\mathbf{1}$ corresponds to a 20-dimensional all-one vector.

To reverse the aforementioned forward process and denoise to generate CDR sequence, predictions need to be made by a neural network $F(\cdot)[j]$, which takes the antibody-antigen context as condition:

$$p\left(s_j^{t-1} \mid \mathcal{R}^t, \mathcal{C}_{ab}, \mathcal{C}_{ag}\right) = \text{Multinomial}\left(F\left(\mathcal{R}^t, \mathcal{C}_{ab}, \mathcal{C}_{ag}\right)[j]\right) \quad (2)$$

As an example, this work uses Diffab (Luo et al., 2022) as the backbone for the generative model to conduct retrieval augmented generation. Note that the proposed retrieval system is generative model agnostic, and the developed modules can be integrated with any diffusion generative model.

## 4 METHODS

We propose RADAb (as demonstrated in Figure 2), a novel structure-informed retrieval-augmented diffusion framework for antibody sequence design and optimization. The model uses a structural retrieval algorithm to search for antibody homologous structures and take their sequences as conditional inputs for the diffusion model to provide homologous patterns and evolutionary information.

### 4.1 STRUCTURAL RETRIEVAL OF CDR FRAGMENTS

The structure of a protein is determined by its sequence, and protein sequences that can fold into similar structures exhibit similar properties. These structurally similar protein sequences contain rich evolutionary information. Based on this, we perform retrieval in the PDB database using CDR structures, aiming to obtain fragments that are similar to the real CDR and have homologous sequences, with the expectation that they possess similar functions.

To balance the quality of results and the retrieval speed, we use MASTER (Zhou & Grigoryan, 2015) for the search. MASTER uses the root-mean-square deviation (RMSD) of backbone atoms as a similarity measure. It queries structural fragments composed of one or more non-contiguous segments and can find all matching fragments from the database within a given RMSD threshold. This allows for fast and accurate searches in the PDB database for protein motifs. Note that MASTER can utilize only the backbone information without any leakage of sequence data during the search process. The retrieval procedure is described in Algorithm 1 and detailed in Appendix A.3.

For the retrieved results, we use the RMSD with the real backbone structure as a score to rank them and filter out the input CDR fragment. For ease of use, we further constructed a CDR-like fragments database (detailed in Appendix A.4). Additionally, to enable the model to learn richer evolutionary information, we filter out identical CDR-like sequences during the training phase. However, to improve the quality of the model's generation, we do not perform similar filters during generation.

---

**Algorithm 1** Structural Retrieval Algorithm Overview

---

1: **Input:** Coordinates set $\mathcal{X} = \{x_k \mid k \in \{1, ..., m\}\}$
2: **Input:** Structure database with $P$ structures $\mathbb{T} = \{\tau_i \mid i \in \{1, ..., P\}\}$
3: Initialize CDR-like fragments set: $\mathbb{A} \leftarrow \emptyset$
4: Initialize structure residues set: $\mathcal{C} \leftarrow \emptyset$
5: Initialize threshold maxA(), maxB(), maxC()
6: **for** $i = 1$ **to** $P$ **do**
7:     $\mathcal{C} \leftarrow$ all residues in $\tau_i$
8:     **for** each residue $j$ in $\mathcal{C}$ **do**
9:         $r \leftarrow RMSD(\mathcal{X}, j)$
10:         **if** $r > \text{maxA}(\mathcal{X})$ **then**
11:             eliminate $j$ from the list $\mathcal{C}$
12:             continue
13:         **end if**
14:         **if** $(r > \text{maxB}(\mathcal{X}))$ **OR** $(cRMSD(\mathcal{X}) > \text{maxC}(\mathcal{X}))$ **then**
15:             continue
16:         **end if**
17:         $A \leftarrow J$
18:         insert match $A$ into $\mathbb{A}$
19:     **end for**
20: **end for**
21: **return** $\mathbb{A}$

---

## 4.2 MODEL ARCHITECTURES

The model takes the antigen-antibody complex's structure and sequence context, along with the sequences of the CDR-like fragments, as conditional inputs to iteratively denoise. The first branch of the model learns the global context information of the complex, while another branch takes the local homologous information of CDR-like fragments as input, aiming to learn the functional similarity and evolutionary information of residues with similar structure. The two branches are combined to generate the antibody CDR sequence jointly.

### 4.2.1 GLOBAL GEOMETRY CONTEXT INFORMATION BRANCH

**Context encoder** A protein is formed by the connection of multiple residues. The features of a single residue mainly include the residue type, backbone atom coordinates, and backbone dihedral angles. The features of each pair of residues mainly include the types of both residues, sequential relative position, spatial distance, and pairwise backbone dihedrals. These features are concatenated and then input into two separate MLPs. The output is denoted as $z_i$ and $y_{ij}$.

**Evolutionary encoder** Recent advances have shown the structure-informed protein language model (PLM) is an excellent tool for creating protein sequence embeddings and providing evolutionary information (Zheng et al., 2023; Shanker et al., 2024). Thus, we take ESM2 (Lin et al., 2023) as an

antibody sequence encoder, aiming to capture the evolutionary relations of antibody residues. The state of antibody sequence with CDR at timestep $t$ is fed into it and output is defined as $e^t$.

**Structure-informed network** The above encoding is used as conditional input to the Structure-informed network. They, along with the CDR sequence and structural state at the current time step, will be input into a stack of Invariant Point Attention (IPA) (Jumper et al., 2021) layers, and jointly transform into a hidden representation $h_i$. Subsequently, the hidden representation $h_i$ is transformed by an MLP to obtain the probability representation $r_{\text{global}}$ of the amino acid type at each CDR site. This probability representation is then input to the local CDR-focused branch.

### 4.2.2 LOCAL CDR-FOCUSED INFORMATION BRANCH

**Post-processing of CDR-like fragments** We first remove the CDR portions from the antibody sequences to obtain the antibody framework sequences. Then, we fill these fragments' short sequences into the antibody framework sequences, thereby constructing a CDR-like sequence matrix $\mathbf{E}$.

**CDR-focused Axial Attention** The local CDR-like branch is constituted of a stack of axial attention layers, referred to as CDR-focused Axial Attention. Given that the CDR-like fragments exhibit structures similar to actual CDRs, we employed a tied row attention mechanism used in MSATransformer (Rao et al., 2021) to leverage these retrieval results. In the standard axial attention (Ho et al., 2019) mechanism, each row and column are considered independently. However, in MSA (Multiple Sequence Alignment), each sequence exhibits relatively similar structural features. Our matrix format is well-suited for adopting a tied row attention mechanism to fully utilize the structural similarity. When calculating the attention scores for each row, this mechanism simultaneously considers the scores of other rows. This approach not only leverages the structural similarity but also reduces memory usage.

The input to CDR-focused Axial Attention is a pseudo-MSA matrix $\mathbf{P}$ in equation 3. The first row of this matrix is initially filled with the antigen-antibody framework sequence, with the CDR region populated by the noisy sequence $R_g^t$ (sampled by $r_{\text{global}}$) at the current time step $t$. From the second row to the k-th row (where k is chosen to be 16, meaning the top 15 retrieved CDR-like sequences are used as conditional input), the rows are filled with the CDR-like sequence matrix $\mathbf{E}$. The constructed matrix $\mathbf{P}$ is then input to CDR-focused Axial Attention to create the homologous embedding and calculate the probability representation $r_{\text{local}}$ (equation 4).

$$\mathbf{P} = \text{concat}\left(\left(S_{ab} \cup R_g^t\right), \mathbf{E}\right) \tag{3}$$

$$
\begin{aligned}
r_{\text{local}}[\cdot, j] &= G_{\text{col}}\left(\mathbf{P}_{\cdot, j}, t\right) \text{ for all } j \in \text{col}, \\
r_{\text{local}}[i, \cdot] &= G_{\text{tiedrow}}\left(\mathbf{P}_{i, \cdot}, t\right) \text{ for all } i \in \text{row}
\end{aligned}
\tag{4}
$$

The row self-attention is computed to capture the internal relationships within the antibody-antigen sequences, while the column self-attention is computed to capture the relationships between the CDR residues and the CDR-like residues.

**Skip connection for information fusion** Although the probability distribution of the CDR region created by the antigen-antibody context features has already been fed into the network, to prevent the loss of antigen-antibody context information during forward propagation, the embedding $r_{\text{local}}$ and $r_{\text{global}}$ are added by a skip connection module (He et al., 2016), then execute *softmax* to obtain the final probability distribution.

### 4.3 MODEL TRAINING AND INFERENCE

**The overall training objective** The training objective is to minimize the probability distributions predicted by the network under two conditions at each time step and the true posterior distribution at the same time step. Therefore, we choose KL divergence between the two distributions at each residue in the CDR region as the training loss function,

$$L_{\text{type}}^t = \mathbb{E}_{\mathcal{R}^t \sim p}\left[\frac{1}{m}\sum_j D_{\text{KL}}\left(q\left(\mathbf{s}_j^{t-1} \mid \mathbf{s}_j^t, \mathbf{s}_j^0\right) \| p\left(\mathbf{s}_j^{t-1} \mid \mathcal{R}^t, \mathcal{C}_{ab}, \mathcal{C}_{ag}, \mathbb{A}\right)\right)\right] \tag{5}$$

The training objective of the whole diffusion process is:

$$L = E_{t \sim \text{ Uniform } (1...T)} L_{\text{type}}^t \tag{6}$$

**Conditional reverse diffusion process** We employ DDPM to generate sequences. The model starts from time step T, initializing each site of the antibody CDR region as a uniform distribution. Then, through the frozen ESM encoder $E(\cdot)$, learned global context network $F(\cdot)[j]$ and the local CDR-focused network $G(\cdot)[j]$, they predict the noise distribution at each time step jointly and denoise step-by-step:

$$e^t = E(S_{ab} \cup R^t) \tag{7}$$

$$\begin{aligned} p\left(s_j^{t-1} \mid \mathcal{R}^t, \mathcal{C}_{ab}, \mathcal{C}_{ag}, \mathbb{A}\right) = \text{Multinomial} \big[ & F\left(\mathcal{R}^t, \mathcal{C}_{ab}, \mathcal{C}_{ag}, e^t\right) \\ & + G\left(F\left(\mathcal{R}^t, \mathcal{C}_{ab}, \mathcal{C}_{ag}, e^t\right), \mathbb{A}\right) \big][j] \end{aligned} \tag{8}$$

During sampling, we remove the CDR region sequences from the antibody structures and fill them with noisy sequence sampled from the uniform distribution. The retrieval process uses the structure of the CDR region as input and outputs a set of CDR-like fragments. Subsequently, this set of CDR-like fragments is fed into the retrieval-augmented diffusion model, serving as a condition along with the antigen-antibody framework context to guide the model in step-by-step denoising and generating the CDR sequences.

# 5 EXPERIMENTS

To evaluate the performance of our model's generation, we utilize two tasks: antibody CDR sequence inverse folding (Section 5.1) and antibody optimization based on sequence design (Section 5.2), to compare with the baselines. Additionally, we conducted ablation experiments and further analysis to demonstrate the effectiveness of the retrieval-augmented method (Section 5.3).

The dataset for training the model is obtained from the SAbDab and our established CDR-like fragments dataset. Following the previous work (Luo et al., 2022), we first eliminated structures with a resolution lower than 4Å and removed antibodies that target non-protein antigens. Chothia (Chothia & Lesk, 1987) in ANARCI (Dunbar & Deane, 2016) is used for renumbering antibody residues. We clustered the SADab datasets based on 50% sequence similarity in the CDR-H3 region, and chose 50 PDB files comprising 63 antibody-antigen complex structures as the test set. To ensure distinct training and test sets, we removed structures from the training set that were part of the same clusters as those in the test set.

## 5.1 ANTIBODY CDR SEQUENCE INVERSE FOLDING

**Baselines** For traditional methods, we simulated a method of grafting using CDR-like data in the process of rational antibody design. Specifically, we directly graft the retrieved top-1 CDR-like fragment onto the antibody framework, termed **Grafting**. For deep learning methods, we selected a series of state-of-the-art protein inverse folding models for comparison with our work, including **ProteinMPNN** (Dauparas et al., 2022), a model that utilizes message passing neural network to design sequences with a fixed protein backbone; **ESM-IF** (Hsu et al., 2022), a protein inverse folding model that trained on millions of predicted structures; **Diffab-fix** (Luo et al., 2022), which can fix the backbone structure and iteratively generate candidate sequences from pure noise in sequence space using diffusion; **AbMPNN** (Dreyer et al., 2023), a model fine-tuned ProteinMPNN on antibody sequence and structure. Because it is not open-sourced, we evaluate it on its own test set. For more baseline details, please refer to Appendix A.5.

**Metrics** To evaluate the accuracy and rationality of the sequences generated by the model, we selected the following three popular evaluation metrics: (1) Amino Acid Recovery (AAR,%): AAR refers to the ratio of positions where the designed sequence and the true CDR sequence have the same amino acid; (2) Self-consistency RMSD (scRMSD, Å ): To calculate scRMSD, we refold the antibody sequences generated by the model using ABodyBuilder2 (Abanades et al., 2023). Then, we align the refolded antibody framework with the original antibody and compute the RMSD of the $C_\alpha$ atoms in the CDR region. (3) Plausibility: We use pseudo-log-likelihood in an antibody language model, AntiBERTy (Ruffolo et al., 2021) to calculate plausibility of the generated sequence.

**Results** As shown in Table 1, RADAb outperforms state-of-the-art methods in each metric and at each CDR region. In particular, in the highly variable and specific CDR-H3 region (Shirai et al., 1999; Raybould et al., 2019), our method achieved a great improvement in AAR compared to best-performing methods Diffab-fix and AbMPNN. The evaluation results indicate that the retrieval-

Table 1: Results of sequence design on SAbDab dataset

| Method | CDR-H1 | | | CDR-H2 | | | CDR-H3 | | |
|---|---|---|---|---|---|---|---|---|---|
| | AAR(%) ↑ | scRMSD ↓ | Plausibility ↑ | AAR(%) ↑ | scRMSD ↓ | plausibility ↑ | AAR(%) ↑ | scRMSD ↓ | plausibility ↑ |
| Grafting | 58.05 | 0.83 | -0.597 | 31.46 | 0.79 | -0.619 | 19.63 | 3.20 | -0.591 |
| ProteinMPNN | 58.58 | 0.64 | -0.603 | 53.18 | 0.61 | -0.568 | 41.77 | 2.27 | -0.605 |
| ESM-IF1 | 53.80 | 0.66 | -0.610 | 46.66 | 0.63 | -0.589 | 29.82 | 2.59 | -0.607 |
| Diffab-fix | 74.93 | 0.66 | -0.512 | 65.41 | 0.59 | -0.532 | 49.17 | 2.24 | -0.541 |
| AbMPNN* | 72.83 | 1.09 | -0.664 | 65.33 | 0.93 | -0.677 | 52.99 | 2.80 | -0.675 |
| **RADAb** | **76.57** | **0.61** | **-0.505** | **66.16** | **0.57** | **-0.530** | **57.02** | **2.23** | **-0.530** |

| Method | CDR-L1 | | | CDR-L2 | | | CDR-L3 | | |
|---|---|---|---|---|---|---|---|---|---|
| | AAR(%) ↑ | scRMSD ↓ | Plausibility ↑ | AAR(%) ↑ | scRMSD ↓ | plausibility ↑ | AAR(%) ↑ | scRMSD ↓ | plausibility ↑ |
| Grafting | 68.53 | 0.85 | -0.506 | 43.19 | 0.52 | -0.573 | 43.61 | 1.08 | -0.395 |
| ProteinMPNN | 45.60 | 0.59 | -0.612 | 46.78 | 0.46 | -0.527 | 47.21 | 0.98 | -0.543 |
| ESM-IF1 | 40.97 | 0.61 | -0.650 | 43.40 | 0.43 | -0.542 | 38.93 | 0.92 | -0.569 |
| Diffab-fix | 79.78 | 0.56 | -0.386 | 81.19 | 0.44 | -0.398 | 67.97 | 0.88 | -0.414 |
| AbMPNN* | 75.06 | 0.73 | -0.543 | 71.63 | 0.56 | -0.528 | 64.51 | 0.91 | -0.544 |
| **RADAb** | **83.72** | **0.54** | **-0.379** | **84.58** | **0.44** | **-0.384** | **73.11** | **0.87** | **-0.384** |

Table 2: Results of long CDRH3 sequence design performance.

| Method | AAR(%) | scRMSD | plausibility |
|---|---|---|---|
| Grafting | 7.79 | 4.05 | -0.785 |
| ProteinMPNN | 46.63 | 2.71 | -0.820 |
| ESM-IF1 | 30.01 | 2.86 | -0.845 |
| Diffab-fix | 42.26 | 3.02 | **-0.740** |
| AbMPNN* | 43.27 | 4.39 | -1.012 |
| **RADAb** | **51.35** | **2.52** | -0.747 |

Table 3: Results of binding energy optimization based on antibody sequence design.

| Method | $\Delta\Delta G\downarrow$ | $\Delta\Delta G$-seq ↓ | IMP-seq(%) ↑ |
|---|---|---|---|
| Grafting | 135.17 | 40.22 | 32.69 |
| ProteinMPNN | 127.14 | 24.72 | 35.51 |
| ESM-IF1 | 162.09 | 42.28 | 33.33 |
| Diffab-fix | 116.36 | 14.05 | 34.52 |
| **RADAb** | **109.16** | **7.06** | **37.30** |

augmented method, by introducing structurally similar homologous sequences, has improved the accuracy, consistency, and rationality of the model's generation.

In addition, CDR-H3 exhibits significant variability in length, sequence, and structure. Typically, deep learning models show decreased performance when generating longer CDR-H3 sequences (Luo et al., 2022; Høie et al., 2024). Therefore, we selected a subset of the test set with CDR-H3 lengths longer than 14 to evaluate the generation performance. As shown in Table 2, while the generation performance of all methods declines to some extent, our method demonstrates consistency and significantly outperforms the others, with a larger margin of improvement.

## 5.2 ANTIBODY FUNCTIONALITY OPTIMIZATION

In this section, we focus on the evolution of antibody sequences and evaluate whether the structure of the evolved sequence has greater functionality compared to the structure of the folded original sequence. To this end, we first fold the designed CDR-H3 sequences with framework sequences and the original real antibody sequences into complete protein structures using ABodyBuilder2. Then, we use *FastRelax* and *InterfaceAnalyzer* in PyRosetta (Alford et al., 2017) to relax the structure and calculate the binding energy $\Delta G$ of the antibody-antigen complex.

**Metrics** We use various metrics to evaluate the efficacy and functionality of our designed antibodies: (1) $\Delta\Delta G$: This metric represents the difference in binding energy between the complex with the designed CDR folded into the structure and the original complex binding energy. (2) $\Delta\Delta G$-seq: This metric measures the difference in binding energy between the complex with the designed CDR sequence folded into the structure and the binding energy of the original antibody sequence folded into the structure. It aims to eliminate errors introduced by the folding tool, allowing for a direct comparison of sequence functionality. (3) IMP-seq: This metric indicates the percentage of designed CDR sequences folded into the structure with a lower (better) binding energy than the original antibody sequences folded into the structure.

**Results** As shown in Table 3, after folding and relaxing, the antibody sequences we designed show a significant decrease in binding energy compared to other methods, and 37.3% of them have better binding energy than those folded from the original antibody sequences.

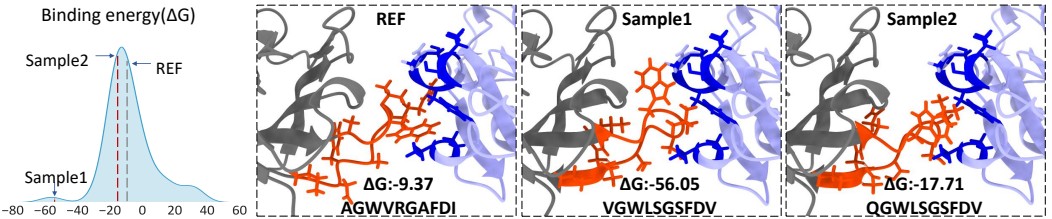

Figure 3: Left: Distribution of the samples' interface energy. Right: Generated CDR-H3 samples and the original structure of PDB: 7d6i antigen-antibody complex. The gray part represents the antibody framework, the red part represents the CDR, and the blue part represents the antigen (with the darker shade indicating the antigen epitope).

Table 4: Ablation Study. $G$ represents using Ground truth as retrieval results, $R$ represents the Retrieval-augment mechanism, and $E$ represents Evolutionary embedding mechanism.

| Ablation | | | AAR(%) | scRMSD | Plausibility |
|---|---|---|---|---|---|
| $G$ | $R$ | $E$ | | | |
| ✔ | ✔ | ✔ | 70.56 | 2.13 | -0.534 |
| ✘ | ✘ | ✔ | 51.36 | 2.23 | -0.538 |
| ✘ | ✔ | ✘ | 52.15 | 2.39 | **-0.529** |
| ✘ | ✘ | ✘ | 49.17 | 2.24 | -0.541 |
| ✘ | ✔ | ✔ | **57.02** | **2.23** | -0.530 |

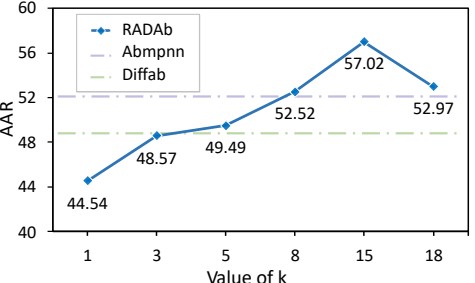

Figure 4: The effect of the value of CDR-like fragments $k$ on model's CDRH3 performance.

To further demonstrate the optimization of antibody sequence functionality by RADAb, we select a specific antigen-antibody complex from the test set (A neutralizing MAb targeting the receptor-binding domain of SARS-CoV-2, PDB: 7d6i). We generate 50 sequences for CDR-H3 and calculate the binding energy $\Delta G$ of the folded structures. Among these, 68% of the samples exhibited lower $\Delta G$ compared to the original complex. As shown in Figure 3, we select two samples as examples. Although they do not achieve the highest AAR, they demonstrate better binding affinity compared to the native structure.

## 5.3 ANALYSIS

**Ablation** We conducted a series of ablation experiments for CDR-H3 following the settings described in Section 5.1 to validate the effectiveness and relative contributions of the additional conditions and data we introduced. The specific objectives are: (1) to verify the effectiveness of the retrieval augment module; (2) to assess the validity of the retrieved data; and (3) to evaluate the effectiveness of the evolutionary embedding.

As shown in Table 4, We demonstrated the retrieval augment module's effectiveness by inputting the CDR sequence's ground truth into this module. We also removed the retrieval augmentation mechanism and the evolutionary embedding mechanism respectively to validate their effectiveness. The experimental results show that both the retrieval augmentation module and the evolutionary embedding module individually improve performance, and using them together maximizes the model's performance.

**Effect of retrieval dataset** To further analyze the benefits brought by the retrieval mechanism and retrieved motifs, we conducted a series of comparative experiments on the value $k$ of CDR-like fragments selected as conditions in the diffusion network, as shown in Figure 4. We found that when the value of $k$ is low, it brings negative benefits to the model, which may be due to overfitting. As the value of $k$ increases, the model performance also gradually improves. When $k$ equals 15, the model achieves the best performance. But when $k$ exceeds 15, the additional information instead introduces noise to the model, leading to performance degradation.

## 6   CONCLUSION AND FUTURE WORKS

In this work, we propose a retrieval-augmented diffusion generative model RADAb for antibody sequence design. This model leverages global geometric information and local template information, incorporating these conditions into the diffusion process to enhance antibody sequence design and optimization. Experimental results demonstrate that RADAb achieves state-of-the-art performance across multiple tasks. The main limitation of this work is that it has not yet been fully validated in wet lab experiments, which will be one of the major tasks in the future. Since we have proposed a comparatively general retrieval method and retrieval-augment framework, another major future task is to extend the model to the design of various protein motifs.

### REPRODUCIBILITY STATEMENT

The code is avalibale at https://github.com/GENTEL-lab/RADAb

### ACKNOWLEDGMENTS

This study has been supported by the National Natural Science Foundation of China [62041209], Natural Science Foundation of Shanghai [24ZR1440600], the Science and Technology Commission of Shanghai Municipality [24510714300].

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

# A    ADDITIONAL DETAILS

## A.1    MODEL DETAILS

For feature dimensions, we set the single residue feature dimension to 128 and the pair feature dimension to 64. We leverage 6 IPA layers to capture geometry information. ESM2 650M is utilized in our model to create the embedding of antibody sequences, and the embedding dimension is 1280. In the local CDR-focus network, two layers of axial attention were used (two tied row self-attention and two column self-attention). The embedding dimension is 384, the hidden dimension is 1536, and number of attention heads is 6.

## A.2    IMPLEMENTATION DETAILS

Our model was developed and executed within the PyTorch framework. For training, We chose the Adam optimizer with a learning rate of 0.0001, weight decay of 0.0, and momentum parameters beta1 and beta2 set to 0.9 and 0.999, respectively. To dynamically adjust the learning rate, we employed plateau as learning rate scheduler. When the validation loss plateaued, the learning rate was reduced by a factor of 0.8, with a minimum learning rate set to 5e-6. The scheduler's patience was set to 10 epochs. The batch size is 8 during training. We design 8 samples for each CDR in the test set. All experiments are run on a single RTX4090 GPU, with a memory storage of 24GB.

Due to the high variability and specificity of the CDRH3 region, and it is considered the most critical part in determining antigen-antibody binding. We conducted separate training for the sequence design of this region, adding and removing noise only for the CDRH3 region in each training iteration, with a total of 100,000 iterations. The other five regions, being more conserved, were trained together for a total of 250,000 iterations (approximately equivalent to 50,000 iterations per region). The reverse generation process time step t is set to 100.

## A.3    IMPLEMENTATION OF STRUCTURAL RETRIEVAL

The input consists of the backbone atom coordinates of each amino acid in the CDR region, forming a set of coordinate points $\mathcal{X}$. $\mathbb{T}$ represents the structures of all proteins in the PDB database. $\mathbb{A}$ represents a set of protein fragments representing CDR-like fragments corresponding to the input CDR structure. $\mathcal{C}$ represents the set of fragments of each structure with a length of m. $J$ represents a linear motif centered on residue $j$ in the structure, with a length equal to the query fragment.

Assume that the coordinates of residue $j$ are aligned with the central residue of $\mathcal{X}$, and then compute the RMSD of $\mathcal{X}$ when aligned onto $\tau$. If the input contains discontinuous multiple structures, cRMSD will be the cumulative RMSD of these structures. MaxA, maxB, and maxC are three different upper-bound thresholds. These thresholds are selected to improve the speed and accuracy of the retrieval algorithm (For detailed proof, please refer to MASTER (Zhou & Grigoryan, 2015)).

## A.4    IMPLEMENTATION OF CDR-LIKE DATABASE CONSTRUCTING

To eliminate the computational overhead caused by structural retrieval during the model's training and inferencing, we followed previous work (Aguilar Rangel et al., 2022) and initially executed the retrieval algorithm on all CDR structures of all antibodies to construct a CDR-like database.

Each of the CDR structures is used as a query to search for structurally similar motifs in the PDB database. The MASTER algorithm is used to match all CDRs against the entire PDB database to find CDR-like structures. This structural search is based on the Kabsch algorithm, using the RMSD of the $C\alpha$ coordinates. For CDR fragments of length 4, the RMSD threshold is 0.4, and the threshold is increased by 0.05 Å for each additional residue (with a maximum threshold set to 1.0 Å). In this way, we obtain a CDR-like fragments database corresponding to all CDR structures. Except for strictly filtering out results identical to real CDR sequences, no CDR sequence information was leaked in this process.

## A.5 BASELINE DETAILS

### A.5.1 TRADITIONAL METHODS

**Rosetta-Fixbb** (Adolf-Bryfogle et al., 2018) Rosetta-Fixbb can use energy functions for antibody CDR sequence design. Since DiffAb has already been proven to outperform it (Luo et al., 2022; Wu & Li, 2023) on sequence design task, we did not conduct additional comparisons.

**Grafting** To simulate the rational design commonly used in traditional antibody design methods, which often involves grafting CDR loops. For each CDR region, we directly selected the top-1 fragment(best) from the retrieval database for the structures in the test set and replaced the corresponding original CDR loop sequences with it.

### A.5.2 DEEP LEARNING METHODS

**ProteinMPNN** (Dauparas et al., 2022) ProteinMPNN is a deep learning framework for protein sequence inverse folding. It leverages a message passing neural network to model the complex relationships between amino acids in a protein structure. We use the antibody's backbone structure as input and keep the sequences outside the CDR regions to be designed fixed. We design sequences for each CDR region separately. The sampling temperature is set to the default value of 0.1.

**Esm-IF1** (Hsu et al., 2022) Esm-IF1 is a protein sequence inverse folding model trained on millions of AlphaFold2 predicted structures. We use the antibody's backbone structure as input and keep the sequences outside the CDR regions to be designed fixed. We design sequences for each CDR region separately. The sampling temperature is set to the value of 0.2.

**Diffab-fix** (Luo et al., 2022) Diffab is a diffusion model that can design sequences of CDR region with a fixed CDR backbone. It takes antigen-antibody framework context as condition to design CDR sequence. For a fair comparison, we retrained it with the default training configuration *fixbb.yml*.

**AbMPNN** (Dreyer et al., 2023) AbMPNN is fine-tuned by antibody structure data and predicted OAS (Observed Antibody Space ) structure data. Its model architecture is consistent with ProteinMPNN but achieves better performance in antibody inverse folding. We use the antibody's backbone structure as input and keep the sequences outside the CDR regions to be designed fixed. However, it is not open-sourced yet, so we evaluate it on its own test set. We design sequences for each CDR region separately. The sampling temperature is set to the default value of 0.1.

## A.6 EXPERIMENT ON CDR-H3'S LENGTH

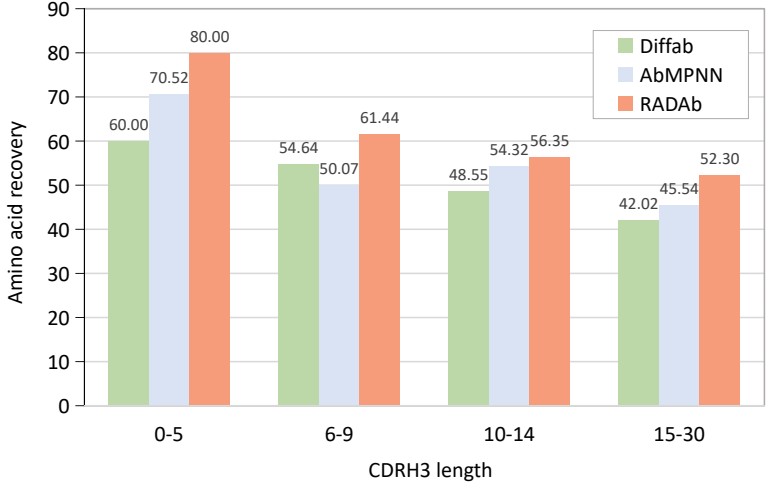

Figure S1: AAR distribution of different CDRH3 length

| Method | $\Delta\Delta G\downarrow$ | $\Delta\Delta G$-seq$\downarrow$ | IMP-seq(%)$\uparrow$ | F-top1$\downarrow$ | F-top2$\downarrow$ | F-top3$\downarrow$ |
|--------|------|----------|-----------|---------|---------|---------|
| Grafting | 135.17 | 40.22 | 32.69 | - | - | - |
| ProteinMPNN | 127.14 | 24.72 | 35.51 | -54.63 | -45.58 | -35.70 |
| ESM-IF1 | 162.09 | 42.28 | 33.33 | -62.65 | -48.81 | -33.86 |
| Diffab-fix | 116.36 | 14.05 | 34.52 | -62.51 | -54.80 | **-46.13** |
| **RADAb** | **109.16** | **7.06** | **37.30** | **-69.30** | **-55.95** | -45.96 |

Table 5: Detailed results of antibody functionality optimization, F means functionality, which refers to $\Delta\Delta G$-seq.

We further evaluated each model's AAR across different CDR-H3 lengths. As shown in Figure S1, although the performance of all models decreases with increasing H3 length, our method still outperforms the others.

### A.7 DETAILED RESULTS OF ANTIBODY FUNCTIONALITY OPTIMIZATION

To further evaluate the model's performance in optimizing antibody functionality, we additionally assessed the $\Delta\Delta G_{seq}$ of the top-1 to top-3 structures generated by the model. The results are shown in Table 5, which further demonstrate that the antibodies optimized by our model exhibit improved functionality.

## B STRUCTURE RECONSTRUCTION

To reconstruct the antibody structure, we use ABodyBuilder2 (Abanades et al., 2023), a deep learning model capable of predicting antibody light chain-heavy chain complexes. It is significantly faster than AlphaFold2 and offers higher prediction accuracy. We insert the designed CDR sequences into the antibody framework sequence, input it into ABodyBuilder2 to fold, and use OpenMM relax to obtain the structure corresponding to the new CDR sequences. Subsequently, we align the structure to the real antibody framework. Finally, we use the *fastrelax* function in PyRosetta (Alford et al., 2017), with the score function set to *ref2015* and max iteration set to 1000, to relax the structure.

## C TRAINING AND INFERENCE ALGORITHM

In this section, we provide a detailed algorithm for the training (Algorithm 2) and inferencing (Algorithm 3) processes.

---
**Algorithm 2** Training Procedure of RADAb
---
1: Coordinates set $\mathcal{X} = \{x_k \mid k \in \{1, ..., m\}\}$
2: $\mathbb{A} = \text{Retrieval}(\mathcal{X})$
3: **while** not convergence **do**
4:      $t \sim \text{Uniform}(1, ..., T)$
5:      $q\left(s_j^{t-1} \mid s_j^t, s_j^0\right) = \frac{q\left(s_j^t \mid s_j^{t-1}\right) \cdot q\left(s_j^{t-1} \mid s_j^0\right)}{q\left(s_j^t \mid s_j^0\right)}$
6:      $S = S_{\text{fr}} \cup s_j^t$
7:      $e^t = E(S)$
8:      Context conditions $\mathcal{C} \leftarrow \{\mathcal{R}^t, \mathcal{C}_{ab}, \mathcal{C}_{ag}\}$
9:      $p\left(s_j^{t-1} \mid \mathcal{C}, \mathbb{A}\right) = \text{Multinomial}\left[F\left(\mathcal{C}, e^t\right) + G\left(F\left(\mathcal{C}, e^t\right), \mathbb{A}\right)\right][j]$
10:      $L_{\text{type}}^t = \mathbb{E}_{\mathcal{R}^t \sim p}\left[\frac{1}{m}\sum_j D_{\text{KL}}\left(q\left(s_j^{t-1} \mid s_j^t, s_j^0\right) \| p\left(s_j^{t-1} \mid \mathcal{C}, \mathbb{A}\right)\right)\right]$
11:      $F\left(\mathcal{C}, e^t\right), G\left(F\left(\mathcal{C}, e^t\right), \mathbb{A}\right) \leftarrow \text{Adam}\left(L_{\text{type}}^t\right)$
12: **end while**
13: **return** $F(\cdot), G(\cdot)$
---

---

**Algorithm 3** Sampling Procedure of RADAb

---

1: $s_j^T \sim \text{Uniform}(20)$
2: Backbone coordinates set $\mathcal{X} = \{x_k \mid k \in \{1, ..., m\}\}$
3: $\mathbb{A} = \text{Retrieval}(\mathcal{X})$
4: **for** $t = T$ **to** 1 **do**
5:     $S = S_{\text{fr}} \cup s_j^t$
6:     $e^t = E(S)$
7:     Context conditions $\mathcal{C} \leftarrow \{\mathcal{R}^t, \mathcal{C}_{ab}, \mathcal{C}_{ag}\}$
8:     $p\left(s_j^{t-1}\right) = \text{Multinomial}\left[F\left(\mathcal{C}, e^t\right) + G\left(F\left(\mathcal{C}, e^t\right), \mathbb{A}\right)\right]$
9:     sample $s_j^{t-1}$ from $p\left(s_j^{t-1}\right)$
10:    $R^{t-1} = \{s_j^{t-1} \mid j \in (a+1, \ldots, a+m)\}$
11: **end for**
12: **return** $R^0$

---

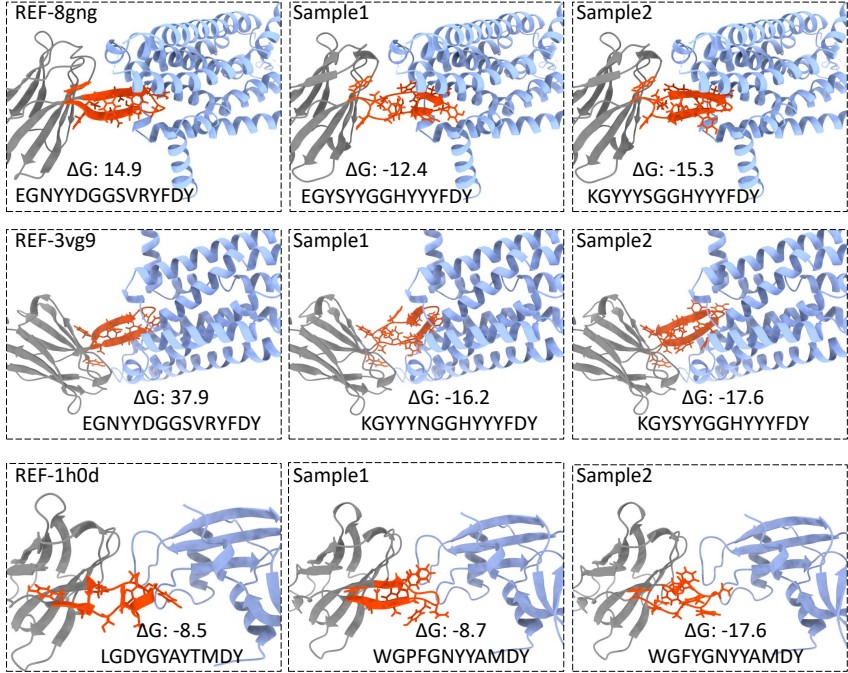

Figure S2: Optimized antibodies with lower binding energy. The gray parts represent the antibody framework, the red parts indicate the designed CDR regions, and the blue parts represent the antigen.

## D    CASE STUDY

We select a portion of the optimized antibodies in Figure S2. They achieved lower binding energy compared to the original antibody structures.

## E    THE MOTIVATION FOR ONLY CONSIDERING SEQUENCE DESIGN

Based on our observations, inverse folding represents a more practical scenario. Current structure-sequence co-design methods typically involve masking the CDR while retaining the presence of an antibody framework backbone, which represents a relatively uncommon use case. In most practical scenarios, we either have access to the full complex structure of the template antibody and the antigen, allowing us to perform inverse folding, or we lack a template molecule entirely, necessitating full atom and *de novo* design. This is also why researches from pharmaceutical companies and efforts involving in vitro experiments on antibody loop regions tend to focus more on designing

antibodies through inverse folding, as evidenced by several recent studies (Shanehsazzadeh et al., 2023a;b; Frey et al., 2023; Høie et al., 2024; Shanker et al., 2024). This rationale underpins our decision to concentrate solely on this aspect in our work.

Another reason is that performing sequence-structure co-design while adhering to our retrieval-based approach would risk data leakage. This is because our retrieval process relies on the known structure of the CDR region, which is only possible when the CDR backbone is already known. At the same time, we recognize that epitope-specific full antibody *de novo* design, including full-atom design, is highly valuable. We are actively developing retrieval systems for PPI (Protein-Protein Interaction) retrieval to avoid potential data leakage and enhance our model.

## F    LIMITATIONS

Due to the inherent limitations of the MASTER algorithms, the retrieved linear motifs may have structural issues. Despite careful screening and filtering, a tiny portion of the data might have lengths that differ from the original CDR loops or may even become discontinuous due to missing residues. These exceptional cases may have a negative impact on our model. We hope that advances in structural retrieval and improvements in alignment will jointly address this issue.

When constructing the CDR-like fragments database, searching the entire PDB database using all CDR structures from the SabDab database takes approximately 100 hours. Additionally, the ESM2 encoding used to capture antibody sequence evolution information and the axial attention focused on local CDR in the denoising network require more computational resources than typical diffusion methods.

It is important to note that applying current retrieval mechanism to structure and sequence co-design poses a significant risk of data leakage. Specifically, the retrieval process inherently relies on using the CDR structure information as a query, which practically transforms the task into one resembling inverse folding. However, one potential approach is to incorporate PPI (Protein-Protein Interaction) retrieval, and we are currently experimenting with MASIF(Gainza et al., 2020; 2023) for this purpose. This will be explored further as part of our future work.

