# OpenReview forum: "Retrieval Augmented Diffusion Model for Structure-informed Antibody Design and Optimization"
_ICLR.cc/2025/Conference — ICLR 2025 Poster_

### Official Review · Reviewer_6nkE · 2024-10-30

**Soundness:** 2
**Presentation:** 3
**Contribution:** 2
**Rating:** 6
**Confidence:** 4

**Summary:**

In this paper authors propose a method to improve upon inverse folding methods by mining PDB for similar structural motifs and use an MSATransformer to take them into account when predicting the loop sequences, along side the usual structure-conditioned diffusion model as used in DiffAb. Inclusion of the mined structures improved the performance of the model in the chosen experiments inverse folding and energy optimization experiments.

**Strengths:**

Inclusion of the data-mined motifs from PDB with similar loop structures is a quite nice and intuitive improvement. That helps the performance of the model. The paper is clearly written and easy to follow. Ablation is performed on the key changes introduced, showing that they are all beneficial.

**Weaknesses:**

Authors state that other models "typically create antibodies from scratch without template constraints, leading to model optimization challenges and poor generative capability". This is a rather harsh critique and I would like such sentences to be backed either by citations or experiments directly showing that. It is true that the proposed approach did generally better than baselines in the chosen experiments, but the difference to DiffAb was marginal and there can be a multitude of reasons for that difference.

In the energy optimization experiment authors say that authors use structures "folded from the original antibody sequences" to evaluate the energy. Folding can be arbitrarilly bad in determining the loop strucutres (esp. CDR H3). So this might very much end up measuring just the agreement between the folding method and the method being tested (how common the folding method finds the sequence to be) instead of how good the energy trully would be. If sequence recovery is reasonably high, one could use grafting and MD to get higher quality/more plausable complexes. Actually, what is the RMSD to original crystal structure, of the reference structure your approach produces (RMSD of original co-crystal vs folding original Ab and runing Rosetta fast relax)?

While in general I like the idea of data mining the PDB for similar loops and using them as reference. I find that the paper is a marginal improvement on existing literature and is of somewhat limited scope for a ICLR paper (only doing inverse folding, while say DiffAb can also do complete loop sequence + structure redesign, and having only two experiments, one of which is not fully convincing to me).

**Questions:**

Did you investigate using a similar approach for structure and sequence re-design (as done by DiffAb)? This would expand the scope of the paper.

When you ablate the numberk of reference loop sequences fed to the model (Figure 4), you show that the optimal number of sequences to use is 15 "But when k exceeds 15, the additional information instead introduces noise to the model, leading to performance degradation.". Do you have any motivation why this would be the case? I'd imagine that as long as the mined loops satisfy some RMSD/quality threshold, they should always be beneficial to the model.

---

> ### Author Response · Authors · 2024-11-20
> **Thank the reviewer for the valuable questions and suggestions. Answer to Weakness and Questions Part1.**
>
> Thank you for your thoughtful and constructive review of our paper. We greatly appreciate your feedback and are pleased to know that you find the work's strength in some aspects. We provide the following response to address your concern.
>
> W1: Authors state that other models "typically create antibodies from scratch without template constraints, leading to model optimization challenges and poor generative capability". This is a rather harsh critique and I would like such sentences to be backed either by citations or experiments directly showing that. It is true that the proposed approach did generally better than baselines in the chosen experiments, but the difference to DiffAb was marginal and there can be a multitude of reasons for that difference.
>
> **A1**: Thanks for pointing this out. We have now revised the sentence to "While these works are undoubtedly powerful, they often generate antibodies from scratch without incorporating explicit structure constraints, which can introduce challenges in designing functional antibodies (Zhou et al., 2024)." accrodingly. This revision has also been highlighted in the updated manuscript.
>
> We fully recognize the significant contributions of these works, while it is true that *de novo* CDR sequence and structure co-design approaches face certain challenges. For example, studies like ABDPO [1] (Table 1) and ProteinBench [2] (Table 6) highlight the inherent limitations of *de novo* methods for designing functional antibody CDRs. In addressing such issues, ABDPO employs energy-based preference optimization, whereas our approach focuses on functionally guided motif retrieval augmentation.
>
> Regarding the similarity between our method and DiffAb, this is primarily because DiffAb serves as the backbone for our model, but our method, which integrates Retrieval-Augmented Generation (RAG) with established biological principles (rational antibody design through grafting, the structure-determines-function paradigm of proteins, and protein co-evolution theories) is generative model agnostic. We selected DiffAb due to its proven performance and controllability in antibody generation tasks.
>
>
>
> W2: In the energy optimization experiment authors say that authors use structures "folded from the original antibody sequences" to evaluate the energy. Folding can be arbitrarilly bad in determining the loop strucutres (esp. CDR H3). So this might very much end up measuring just the agreement between the folding method and the method being tested (how common the folding method finds the sequence to be) instead of how good the energy trully would be. If sequence recovery is reasonably high, one could use grafting and MD to get higher quality/more plausable complexes. Actually, what is the RMSD to original crystal structure, of the reference structure your approach produces (RMSD of original co-crystal vs folding original Ab and runing Rosetta fast relax)?
>
> **A2:**
>
> Good points, folding can indeed be challenging when determining loop structures, especially CDR-H3. However, previous state-of-the-art inverse folding methods [3] have employed ABodyBuilder2 [4], an effective tool for antibody folding, to mitigate systematic errors. We have adopted this strategy for structure prediction, and our method has demonstrated superior performance compared to previous approaches. The average RMSD between the original co-crystal and the folded original antibody, using Rosetta fast relax with ABodyBuilder2 for CDR-H3, is 0.55 Å. This result indicates that ABodyBuilder2 is a reliable tool for this purpose.

---

> ### Author Response · Authors · 2024-11-20
> **Thank the reviewer for the valuable questions and suggestions. Answer to Weakness and Questions Part2.**
>
> W3: While in general I like the idea of data mining the PDB for similar loops and using them as reference. I find that the paper is a marginal improvement on existing literature and is of somewhat limited scope for a ICLR paper (only doing inverse folding, while say DiffAb can also do complete loop sequence + structure redesign, and having only two experiments, one of which is not fully convincing to me).
>
> **A3:**
>
> Thanks for your kind words! Below, we provide clarifications on this point to address your concerns:
>
> 1. Based on our observations, inverse folding represents a more practical scenario. Current structure-sequence co-design methods typically involve masking the CDR while retaining the presence of an antibody framework backbone, which represents a relatively uncommon use case. In most practical scenarios, we either have access to the full complex structure of the template antibody and the antigen, allowing us to perform inverse folding, or we lack a template molecule entirely, necessitating full atom and *de novo* design. This is also why research from pharmaceutical companies and efforts involving in vitro experiments on antibody loop regions tend to focus more on designing antibodies through inverse folding, as evidenced by several recent studies [5][6][7][8][9]. This rationale underpins our decision to concentrate solely on this aspect in our work. We will include this discussion in the revised manuscript.
>
> 2. Another reason is that performing sequence-structure CDR co-design while adhering to our retrieval system would risk data leakage. This is because our retrieval process relies on the known structure of the CDR region, which is only possible when the structure is already known. At the same time, we recognize that epitope-specific full antibody *de novo* design, including full-atom design, is highly valuable. Unfortunately, we have not yet developed a retrieval system to enhance the resolution of such problems. One potential approach is PPI (Protein-Protein Interaction) retrieval, which we are actively exploring as part of our future work.
>
> 3. We would also like to clarify that RADAb is the first work to introduce RAG into antibody design. We believe this is one of the key contributions of our work, providing significant inspiration for the application of RAG techniques in protein design.
>
> We will include and highlight this discussion in appendix of the revised manuscript.

---

> ### Author Response · Authors · 2024-11-20
> **Thank the reviewer for the valuable questions and suggestions. Answer to Weakness and Questions Part3.**
>
> Q1: Did you investigate using a similar approach for structure and sequence re-design (as done by DiffAb)? This would expand the scope of the paper.
>
> **A4:**
> It is important to note that applying current retrieval mechanism to structure and sequence co-design poses a significant risk of data leakage. Specifically, the retrieval process inherently relies on using the CDR structure information as a query, which practically transforms the task into one resembling inverse folding. However, one potential approach is to incorporate PPI (Protein-Protein Interaction) retrieval, and we are currently experimenting with MASIF [10][11] for this purpose. This direction will be explored further as part of our future work.
>
> For this reason, our future focus is mainly on epitope-specific full antibody *de novo* design, where we assume that the complete antibody framework template is unavailable.
> The core idea of this retrieval method is to use the antigen epitope as input for the retrieval system, retrieving a set of protein fragments capable of interacting with the antigen. These fragments are then used to assist in antibody generation. We believe this represents a meaningful step toward addressing the challenges in *de novo* antibody design while mitigating the risk of data leakage in retrieval-based methods and we are working on it.
>
>
> Q2: When you ablate the numberk of reference loop sequences fed to the model (Figure 4), you show that the optimal number of sequences to use is 15 "But when k exceeds 15, the additional information instead introduces noise to the model, leading to performance degradation.". Do you have any motivation why this would be the case? I'd imagine that as long as the mined loops satisfy some RMSD/quality threshold, they should always be beneficial to the model.
>
> **A5:** This phenomenon could be attributed to the signal-to-noise ratio (SNR). When *k* is set to 15, the model may achieve the optimal SNR. This is because, even when sequences meet the RMSD threshold, as RMSD increases, some sequences may exhibit significant variations. These substantial differences in sequence patterns introduce more noise than benefit to the model, reducing the overall SNR of the data and ultimately leading to poorer generative performance. Additionally, we believe this might be related to our pre-screening method for CDR-like fragments. In the future, we plan to explore more advanced retrieval strategies to improve the performance.
>
> Thanks again for the review. We hope the above explanation will address your concerns and encourages you to consider increasing the score.

---

> ### Author Response · Authors · 2024-11-20
> **Thank the reviewer for the valuable questions and suggestions. Answer to Weakness and Questions Part4.**
>
> Reference:
>
> [1] Zhou, X., Xue, D., Chen, R., Zheng, Z., Wang, L., \& Gu, Q. (2024). Antigen-Specific Antibody Design via Direct Energy-based Preference Optimization. arXiv preprint arXiv:2403.16576.
>
> [2] Ye, F., Zheng, Z., Xue, D., Shen, Y., Wang, L., Ma, Y., ... \& Gu, Q. (2024). ProteinBench: A Holistic Evaluation of Protein Foundation Models. arXiv preprint arXiv:2409.06744.
>
> [3] Dreyer, F. A., Cutting, D., Schneider, C., Kenlay, H., \& Deane, C. M. (2023). Inverse folding for antibody sequence design using deep learning. arXiv preprint arXiv:2310.19513
>
> [4] Abanades, B., Wong, W. K., Boyles, F., Georges, G., Bujotzek, A., \& Deane, C. M. (2023). ImmuneBuilder: Deep-Learning models for predicting the structures of immune proteins. Communications Biology, 6(1), 575.
>
> [5] Shanehsazzadeh, A., Bachas, S., McPartlon, M., Kasun, G., Sutton, J. M., Steiger, A. K., ... \& Meier, J. (2023). Unlocking de novo antibody design with generative artificial intelligence. bioRxiv, 2023-01.
>
> [6] Shanehsazzadeh, A., Alverio, J., Kasun, G., Levine, S., Khan, J. A., Chung, C., ... \& Bachas, S. (2023). In vitro validated antibody design against multiple therapeutic antigens using generative inverse folding. bioRxiv, 2023-12.
>
> [7] Frey, N. C., Berenberg, D., Zadorozhny, K., Kleinhenz, J., Lafrance-Vanasse, J., Hotzel, I., ... \& Saremi, S. Protein Discovery with Discrete Walk-Jump Sampling. In The Twelfth International Conference on Learning Representations.
>
> [8] Høie, M., Hummer, A., Olsen, T., Nielsen, M., \& Deane, C. (2023). AntiFold: Improved antibody structure design using inverse folding. In NeurIPS 2023 Generative AI and Biology (GenBio) Workshop.
>
> [9] Shanker, V. R., Bruun, T. U., Hie, B. L., \& Kim, P. S. (2024). Unsupervised evolution of protein and antibody complexes with a structure-informed language model. Science, 385(6704), 46-53.
>
> [10] Gainza, P., Sverrisson, F., Monti, F., Rodola, E., Boscaini, D., Bronstein, M. M., \& Correia, B. E. (2020). Deciphering interaction fingerprints from protein molecular surfaces using geometric deep learning. Nature Methods, 17(2), 184-192.
>
> [11] Gainza, P., Wehrle, S., Van Hall-Beauvais, A., Marchand, A., Scheck, A., Harteveld, Z., ... \& Correia, B. E. (2023). De novo design of protein interactions with learned surface fingerprints. Nature, 617(7959), 176-184.

---

> ### Comment · Reviewer_6nkE · 2024-11-26
>
> Thank you for your rebuttal. I've read your answers and other reviews and will maintain my score.

---

> > ### Author Response · Authors · 2024-11-26
> >
> > We greatly appreciate your recognition of our paper's strengths. Thank you again for taking the time to review our work and providing detailed comment for helping us to improve the paper!

---

### Official Review · Reviewer_Fxue · 2024-11-02

**Soundness:** 3
**Presentation:** 3
**Contribution:** 2
**Rating:** 6
**Confidence:** 3

**Summary:**

This paper introduces a retrieval-augmented diffusion framework for anti-body design and optimization. Its key contribution is that it retrieves structurally-similar protein segments to the CDR, and use them as conditions in the diffusion generation. Experiments demonstrates impressive results, showing improved performances in antibody CDR sequence inverse folding and antibody functionality optimization. Ablation experiments show that both the retrieval augmentation and evolutionary embedding (ESM2) contribute to the improved performance.

**Strengths:**

Originality: See the weaknesses below

Quality: The paper overall has good quality. The experiment is sound.

Clarity: The paper is written clearly.

Significance: The paper significantly improves upon the baseline models in terms of the metrics. It has good significance.

**Weaknesses:**

Originality: There has been retrieval-augmented diffusion model proposed in other applications, such as in [1] and [2] (protein-specific 3D molecule generation). Although the paper is the first to introduce it in the antibody design, the method novelty is fair due to the prior works. The paper needs to also cite previous retrieval-augmented generative models, and clearly state the contribution in terms of the method.


[1] Blattmann, Andreas, et al. "Retrieval-augmented diffusion models." Advances in Neural Information Processing Systems 35 (2022): 15309-15324.

[2] Huang, Zhilin, et al. "Interaction-based Retrieval-augmented Diffusion Models for Protein-specific 3D Molecule Generation." Forty-first International Conference on Machine Learning.

**Questions:**

N/A

---

> ### Author Response · Authors · 2024-11-20
> **Thank the reviewer for the valuable questions and suggestions. Answer to Weakness and Questions Part1.**
>
> Thank you for your thoughtful and constructive review of our paper. We greatly appreciate your feedback and are pleased to know that you find the work's strength in quality, clarity and significance.We provide the following response to address your concern.
>
> W1:
> Originality: There has been retrieval-augmented diffusion model proposed in other applications, such as in [1] and [2] (protein-specific 3D molecule generation). Although the paper is the first to introduce it in the antibody design, the method novelty is fair due to the prior works. The paper needs to also cite previous retrieval-augmented generative models, and clearly state the contribution in terms of the method.
>
> [1] Blattmann, Andreas, et al. "Retrieval-augmented diffusion models." Advances in Neural Information Processing Systems 35 (2022): 15309-15324.
>
> [2] Huang, Zhilin, et al. "Interaction-based Retrieval-augmented Diffusion Models for Protein-specific 3D Molecule Generation." Forty-first International Conference on Machine Learning.
>
> **A1**:
> We fully acknowledge the success of many approaches in other applications, which have been a significant source of inspiration for our work. To clarify, **both** the references you mentioned—[1] Blattmann et al. and [2] Huang et al.—have been already cited in our initial manuscript, specifically in the sections titled 'Diffusion Generative Models' and 'Retrieval-Augmented Generative Models' in Section 2. These citations will be highlighted in the revised version of the manuscript, and we will reconstruct the Related work section to further discuss the application to high-quality generation of retrieval-augmented models and diffusion models.
>
> While we recognize the importance of prior work in the field, our contribution lies in first adapting these kind of method specifically for antibody and protein design—a domain with unique challenges and requirements not addressed in the referenced studies. Additionally, we integrate Retrieval-Augmented Generation (RAG) with established biological principles, such as rational antibody design through grafting, the structure-determines-function paradigm of proteins, and protein co-evolution theories. This combination of biological knowledge with modern AI techniques like RAG ensures that our approach is both innovative and well-motivated.
>
> We hope this clarifies the issue, and we are happy to make any further adjustments to improve the clarity of our contribution and originality. Thanks again for the review. We hope the above explanation will address your concerns and encourages you to consider increasing the score.

---

> > ### Comment · Reviewer_Fxue · 2024-11-26
> >
> > Thanks the authors for the response! After reading the rebuttal and the discussions with other reviewers, I maintain my rating.

---

> > > ### Author Response · Authors · 2024-11-26
> > >
> > > We greatly appreciate your recognition of our paper's strengths. Thank you again for taking the time to review our work and providing detailed comment for helping us to improve the paper!

---

### Official Review · Reviewer_vhAX · 2024-11-04

**Soundness:** 2
**Presentation:** 2
**Contribution:** 3
**Rating:** 5
**Confidence:** 3

**Summary:**

This paper combines the popular technique RAG with antibody design by extracting templates from PDB to enrich evolutionary information, leading to better generalization ability. To be specific, the extracted fragments are based on backbone structures and some properties using a novel retrieval mechanism to utilize both structural and evolutionary information. Experiments demonstrate the effectiveness of the proposed RADAb framework.

**Strengths:**

1. I really like the idea of incorporating RAG into antibody design in such an elegant manner, which not only fulfills the relatively blank space of known data but also utilizes evolutionary information. This is intuitive and turned out to be effective.
2. Nice illustrations, which clearly convey the architecture and designs of authors.

**Weaknesses:**

1. The experiment part is relatively weak. For example, for antibody functionality optimization task, only one simple experiment is provided.
2. Baselines are inadequate. Large efforts have been devoted to studying antibody design, such as [1][2][3]:. They utilize the same experimental settings and metrics, which should be taken into consideration.
3. If I comprehend correctly, although RADab is capable of capturing structural and sequential information, it is still a sequence design work, rather than sequence-structure co-design or even full-atom generation work, which is a major weakness compared to other methods.

[1]Kong X, Huang W, Liu Y. End-to-end full-atom antibody design[J]. arXiv preprint arXiv:2302.00203, 2023.

[2]Xiangzhe Kong, Wenbing Huang, and Yang Liu. Conditional antibody design as 3d equivariant
graph translation. arXiv preprint arXiv:2208.06073, 2022.

[3]Shitong Luo, Yufeng Su, Xingang Peng, Sheng Wang, Jian Peng, and Jianzhu Ma. Antigen-specific
antibody design and optimization with diffusion-based generative models for protein structures.
Advances in Neural Information Processing Systems, 35:9754–9767, 2022.

**Questions:**

1. Is the provided experimental results correct? According to previous works, can $\Delta \Delta G$ be optimized on such a huge scale?  (~100kcal/mol?) Even for the selected examples in 5.2, $\Delta \Delta G$ only changed like ~-40, far away form claimed 109.16.
2. Also, $\Delta \Delta G$ is usually expressed using negative numbers to demonstrate lower energy is achieved. Why $\Delta \Delta G$ is positive in Tab. 3? Suppose you are using the absolute value of $\Delta \Delta G$ , since lower $\Delta G$ is better, why lower $|\Delta \Delta G|$ indicates better performance? Please explain this.

---

> ### Author Response · Authors · 2024-11-20
> **Thank the reviewer for the valuable questions and suggestions. Answer to Weakness and Questions Part1.**
>
> Thank you for your thoughtful and constructive review of our paper. We greatly appreciate your feedback and are pleased to know that you find the work's strength in the idea and illustrations. Upon review, it seems there may have been a misunderstanding, which could have contributed to the lower score.
> We provide the following response to address your concern.
>
> W1: The experiment part is relatively weak. For example, for the antibody functionality optimization task, only one simple experiment is provided.
>
> **A1**: Our experimental setup aligns with established antibody design studies that typically involve two types of experiments: one for designing the CDR regions and the other for optimizing the antibody[1][2]. We acknowledge that the scope of our experiments is not as broad as those focusing on antibody sequence-structure co-design. This is because our work focuses on structure-conditioned antibody sequence design, also known as inverse folding. Unlike co-design approaches, our method assumes that the antibody CDR sequence is unknown and design it using a fixed structure. This strategy has been proven to be highly effective in previous works [3][4][5][6][7], which have demonstrated that designing sequences by fixing the structure of proteins can enhance the functionality of antibodies.
>
> W2: Baselines are inadequate. Large efforts have been devoted to studying antibody design, such as [1][2][3]:. They utilize the same experimental settings and metrics, which should be taken into consideration.
> [1]Kong X, Huang W, Liu Y. End-to-end full-atom antibody design[J]. arXiv preprint arXiv:2302.00203, 2023.
> [2]Xiangzhe Kong, Wenbing Huang, and Yang Liu. Conditional antibody design as 3d equivariant graph translation. arXiv preprint arXiv:2208.06073, 2022.
> [3]Shitong Luo, Yufeng Su, Xingang Peng, Sheng Wang, Jian Peng, and Jianzhu Ma. Antigen-specific antibody design and optimization with diffusion-based generative models for protein structures. Advances in Neural Information Processing Systems, 35:9754–9767, 2022.
>
> **A2**: Thank you for your feedback. As stated in Figure 1C of the paper, our goal is to design better antibody sequence by inverse folding to achieve superior functional performance, rather than generating entirely new antibodies. This distinction makes our task different from typical sequence-structure co-design tasks, and therefore, a direct comparison with works like MEAN[8], DyMEAN[9], and AbX[2], which focus on antibody CDR structure-sequence co-design or *de novo* design, is not appropriate. Instead, we compared our approach with the fixed-structure version of DiffAb and other state-of-the-art antibody inverse folding works, which is more aligned with our task and allows for a more meaningful and fair comparison.
>
> W3: If I comprehend correctly, although RADab is capable of capturing structural and sequential information, it is still a sequence design work, rather than sequence-structure co-design or even full-atom generation work, which is a major weakness compared to other methods.
>
> **A3**: You are correct that our work currently focuses solely on sequence design. Below, we provide clarifications on this point to address your concerns:
> 1. Based on our observations, inverse folding represents a more practical scenario. Current structure-sequence co-design methods typically involve masking the CDR while retaining the presence of an antibody framework backbone, which represents a relatively uncommon use case. In most practical scenarios, we either have access to the full complex structure of the template antibody and the antigen, allowing us to perform inverse folding to design sequence, or we lack a template molecule entirely, necessitating full atom and *de novo*  design. This is also why researches from pharmaceutical companies and efforts involving in vitro experiments on antibody loop regions tend to focus more on designing antibodies through inverse folding, as evidenced by several recent studies [3][4][5][6][7]. This rationale underpins our decision to concentrate solely on this aspect in our work.
> 2. Another reason is that performing sequence-structure co-design while adhering to our retrieval-based approach would risk data leakage. This is because our retrieval process relies on the known structure of the CDR region, which is only possible when the CDR backbone is already known. At the same time, we recognize that epitope-specific full antibody *de novo* design, including full-atom design, is highly valuable. We are actively developing retrieval systems for PPI (Protein-Protein Interaction) retrieval to avoid potential data leakage and enhance our model.
> 3. We would also like to clarify that RADAb is the first work to introduce RAG into antibody design. We believe this is one of the key contributions of our work, providing significant inspiration for the application of RAG techniques in protein design.
> We will include and highlight this discussion in appendix of the revised manuscript.

---

> > ### Comment · Reviewer_vhAX · 2024-11-20
> >
> > Thanks for the comprehensive reply. In case your model is trained on SAbDab dataset, in which the sequence is known, you can retrieve templates and filter out similar ones to guarantee no data leakage. AIso, I am confused as to why co-design can risk data leakage; if the retrieved fragments are precisely CDR regions, that should be called data leakage.

---

> > > ### Author Response · Authors · 2024-11-21
> > > **Thanks to the reviewer for the comment**
> > >
> > > Thanks for your response. We did implement filtering to ensure that similar retrieval results do not lead to any data leakage. Regarding the co-design task, the goal of co-design is to simultaneously design a sequence and a matching structure, while sequence design focuses on inverse folding given a known CDR structure. In our approach, the retrieval process assumes that the CDR structure is known, and the input to the model's retriever is the backbone structure of the CDR region with an unknown sequence. However, in co-design tasks, the backbone structure of the CDR region is assumed to be unknown, making it impossible to use the CDR backbone structure as input for retrieval. This limitation prevents retrieval-based methods from being applied directly to co-design tasks.

---

> > > > ### Comment · Reviewer_vhAX · 2024-11-21
> > > >
> > > > Co-design should contain both structural and sequential information, rather than "the structure of the CDR region is assumed to be unknown." Since your dataset gives both the structure and sequence of the wild type, I don't see why this cannot be achieved. It is not necessarily that RAG must be used to enhance only the sequence part. You can still use similar templates of structure and sequence to guide the generation. Intuitively, with the guidance of both information,  it should strengthen the performance. This seems not to be a proper excuse for only designing sequences. Also, I noticed that in your figure 2B,  when the sequence of CDR has changed, the corresponding structure stays still. Although your work does not involve structure design, this could be misleading as well.

---

> > > > > ### Author Response · Authors · 2024-11-21
> > > > > **Thanks to the reviewer for the comment**
> > > > >
> > > > > We understand that this point can indeed be confusing. Here is a more detailed explanation:
> > > > >
> > > > > Firstly, let's clearly define the concept of co-design as outlined in DiffAb[1], MEAN[2], AbX[3]... In inference process of co-design, the process involves masking both the sequence and structure of the CDR region given a template antibody framework, allowing the model to generate both the sequence and structure simultaneously. In other words, the goal of co-design is to generate CDRs without any prior knowledge of the template CDR's sequence and structure.
> > > > >
> > > > > Our retrieval system, however, is based on structural retrieval using the template CDR motif backbone. The underlying logic is to match the template antibody CDR structures to CDR-like fragments. In this context, we must utilize the structural information of the template's CDR. This requirement conflicts with the precondition of co-design, which assumes complete ignorance of the template CDR structure and sequence. We cannot pretend to be unaware of the template CDR structure, which is where the potential for information leakage arises.
> > > > >
> > > > > Considering this, the reason we state that it is challenging to apply RAG in CDR co-design is precisely because we should not have any prior knowledge of the template CDR information. For co-design, this constraint prevents us from using any structural or sequence information of the antibody CDR for retrieval. To the best of our knowledge, the only non-leaking strategy is to use the antigen epitope information to search for its corresponding binding partner, but currently, there is no public available system to achieve this and we are working on this.
> > > > >
> > > > > We agree with your observation regarding Figure 2B being potentially misleading, and we have revised the figure and it's caption accordingly in the new version of our manuscript.
> > > > >
> > > > > We truly appreciate your constructive feedback, which has helped us improve the clarity of our work. We hope this explanation further clarifies our approach and contributes positively to your evaluation.
> > > > >
> > > > > [1] Luo, S., Su, Y., Peng, X., Wang, S., Peng, J., & Ma, J. (2022). Antigen-specific antibody design and optimization with diffusion-based generative models for protein structures. Advances in Neural Information Processing Systems, 35, 9754-9767.
> > > > >
> > > > > [2] Kong, X., Huang, W., & Liu, Y. Conditional Antibody Design as 3D Equivariant Graph Translation. In The Eleventh International Conference on Learning Representations.
> > > > >
> > > > > [3] Zhu, T., Ren, M., & Zhang, H. Antibody Design Using a Score-based Diffusion Model Guided by Evolutionary, Physical and Geometric Constraints. In Forty-first International Conference on Machine Learning

---

> > > > > > ### Comment · Reviewer_vhAX · 2024-11-23
> > > > > >
> > > > > > It seems much clearer now. As long as your paper is entitled "Antibody Design and Optimization," your core contribution is to apply RAG to enhance antibody design, just like the inverse folding problem. With the progress of a highly accurate antibody design method, which I believe is powerful concerning antibody design, you found out the designed antibody sequences show general improvement compared to other inverse folding methods. However, I am curious about how this RAG framework can help to improve antibodies, as shown in your title. What is the performance ( $\Delta \Delta G$ drop) based on your top-1 candidates? Can your work surpass or be comparable to other antibody optimization methods? Since in the real application scene, we only focus on several top ones, rather than calculating $\Delta \Delta G$ for sequences, most of which are less stable than the wild types. If the authors can provide such experiments or analysis, I will consider raising my score.
> > > > > > I would also like to express my gratitude for the authors' patience.

---

> > > > > > > ### Author Response · Authors · 2024-11-25
> > > > > > > **Thanks to the reviewer for the comment. We have implemented additional experiments.**
> > > > > > >
> > > > > > > Thanks for pointing this out. We acknowledge that the results presented in Section 5.2 were limited to average values across multiple candidates. We fully agree with your suggestion to compare results based on the top-1 candidate.
> > > > > > >
> > > > > > > Accordingly, we have added relevant experiments which include results of top-1 to top-3 candidates. We have also enriched the content of Table 3, now included in Appendix F of the revised manuscript. The modified table is provided below:
> > > > > > >
> > > > > > > | Method      | $\Delta\Delta G$ | $\Delta\Delta G-seq$ | $IMP-seq(\\%)$ |$\Delta\Delta G-seq(top1)$ | $\Delta\Delta G-seq(top2)$ | $\Delta\Delta G-seq(top3)$ |
> > > > > > > | ----------- | ----------- | ----------- | ----------- |----------- |----------- | ----------- |
> > > > > > > | Grafting      | 135.17       |40.22       |32.69       |-      |-       |-       |-         |
> > > > > > > | ProteinMPNN   | 127.14        |24.72       |35.51       |-54.63      |-45.58       |-35.70      |
> > > > > > > | ESM-IF1   | 162.09        |42.28       |33.33       |-62.65       |-48.81      |-33.86       |
> > > > > > > | DiffAb-fix   | 116.36        |14.05       |34.52       |-62.51       |  -54.80     |**-46.13**       |
> > > > > > > | RADAb   | **109.16**        |**7.06**       |**37.30**       |**-69.30**      |**-55.95**       |-45.96       |
> > > > > > >
> > > > > > >
> > > > > > > We sincerely appreciate your constructive feedback, which has been pivotal in helping us refine our work and present it more clearly and effectively.

---

> > > > > > > > ### Comment · Reviewer_vhAX · 2024-11-25
> > > > > > > >
> > > > > > > > Thanks for the clarification. I have raised my score

---

> > > > > > > > > ### Author Response · Authors · 2024-11-26
> > > > > > > > >
> > > > > > > > > Thanks for your patience and for helping improve the paper! If there’s anything else we can do to further improve the quality, please let us know.

---

> ### Author Response · Authors · 2024-11-20
> **Thank the reviewer for the valuable questions and suggestions. Answer to Weakness and Questions Part2.**
>
> Q1: Is the provided experimental results correct? According to previous works, can $\Delta\Delta G$ be optimized on such a huge scale? (~100kcal/mol?) Even for the selected examples in 5.2, only changed like ~-40, far away from claimed 109.16.
>
> A4: Thank you for your question. The provided experimental results are correct. We kindly point out that there seems to be a misunderstanding regarding the $\Delta\Delta G$ optimization. Below, we provide clarifications on this point to address your concerns:
>
> 1. Definition of $\Delta\Delta G$:
>
>    $\Delta\Delta G = \Delta G_{gen} - \Delta G_{ref} $
>
>    Here, $\Delta G_{gen}$ represents the binding energy of the designed antibody, while $\Delta G_{ref}$ corresponds to the binding energy of the reference (real) antibody. Ideally, binding energy should be less than 0. However, due to inaccuracies in the energy calculation software rosetta, both values may occasionally be computed as higher than 0. The other reason is that the structural clashes between CDR and the antigen could lead to the unreasonable high CDR-Ag $\Delta G$. Because of the complexity of protein interactions and errors of rosetta, it is not possible that every generated antibody will have a negative $\Delta G$, which is appeared in another antibody design approach ABDPO [10], where a detailed discussion on antibody energy calculations was conducted.
>
> 2. Performance Metrics in Prior Works: In previous studies, antibody optimization results are often evaluated using the Improvement Percentage (IMP)[1][2], which measures the percentage of optimized antibodies where $\Delta\Delta G < 0$. In our work, this metric shows a noticeable improvement over previous methods. However, it is important to note that a significant proportion of antibodies still fail during optimization, resulting in $\Delta\Delta G > 0$. When averaged, this leads to an overall positive $\Delta\Delta G$ value, but smaller values are still preferable. This phenomenon has also appeared in other works [10][11][12].
>
> 3. Reducing Folding Errors for Fair Comparison: To reduce folding-related errors and provide a more direct comparison of the designed antibody sequences relative to real antibody sequences, we use ABodyBuilder2 to fold both the designed and reference sequences and compared their binding energy in the folded structures. This approach minimizes discrepancies and enhances fairness. Under this setup, the average $\Delta\Delta G_{seq}$—defined as:
>
>    $ \Delta\Delta G_{seq} = \Delta G_{gen-seq} - \Delta G_{ref-seq} $
>
>    This metric shows a 50% reduction compared to the state-of-the-art method. We believe this represents a significant improvement.
>
> Q2: Also, $\Delta\Delta G$ is usually expressed using negative numbers to demonstrate lower energy is achieved. Why $\Delta\Delta G$ is positive in Tab. 3? Suppose you are using the absolute value of $\Delta\Delta G$ , since lower is better, why lower
> $\Delta\Delta G$ indicates better performance? Please explain this.
>
> **A5**: As detailed in A4, beacuse of the complexity of protein interactions and errors of *rosetta*, it is not possible that every generated antibody will have a negative $\Delta\Delta G$. After averaged, this leads to an overall positive $\Delta\Delta G$ value, but smaller values are still preferable. And compared to others, RADAb's improvement is significant.
>
> Thanks again for the review. We hope the above explanation will address your concerns and encourages you to consider increasing the score.

---

> > ### Comment · Reviewer_vhAX · 2024-11-20
> >
> > For example, in [1] Sec 5.3, the average for all optimized antibodies leads to a lower $\Delta$G, does this imply your method has significantly lower performance regarding failures? In other words, more unstable? (Q1)
> >
> > Based on your description in A5, $\Delta\Delta G_{seq}$ is supposed to be lower, suggesting better performance. However, what I comprehend is that $\Delta G_{ref-seq}$ is fixed, $\Delta G_{gen-seq}$ is expected to have lower energy so it is more stable. Consequently, $\Delta\Delta G = \Delta G_{ref-seq}-\Delta G_{gen-seq}$ is expected to be higher to show its superiority? I am still confused about this setting. (Q2)
> >
> >
> > Thanks again.
> >
> >  [1]Kong X, Huang W, Liu Y. End-to-end full-atom antibody design[J]. arXiv preprint arXiv:2302.00203, 2023.

---

> > > ### Author Response · Authors · 2024-11-21
> > > **Thanks to the reviewer for the comment**
> > >
> > > DyMEAN[1] generates multiple candidates for each antibody and evaluates $\Delta\Delta G$ based on the top-1 candidate ('For each antibody in the test set, we generate 100 candidates and record the ∆∆G of the top-1 candidate'). In contrast, more recent works [2][3][4], including ours, generate multiple candidates and calculate the average $\Delta\Delta G$ across all candidates, leading to differences in the reported results.
> > >
> > > We sincerely apologize for the misunderstanding. In the initial version of our rebuttal, there was a typo in the definition of $\Delta\Delta G_{seq}$ in Part 2, where the order of *gen* and *ref* was mistakenly reversed. We corrected this error and apologize again for the misleading caused by the typo.
> > >
> > > In summary, under the same setup, both $\Delta\Delta G$ and $\Delta\Delta G_{seq}$ follow the same principle: the smaller, the better.
> > >
> > > [1] Kong, X., Huang, W., & Liu, Y. (2023, July). End-to-End Full-Atom Antibody Design. In International Conference on Machine Learning (pp. 17409-17429). PMLR.
> > >
> > > [2] Dreyer, F. A., Cutting, D., Schneider, C., Kenlay, H., & Deane, C. M. (2023). Inverse folding for antibody sequence design using deep learning. arXiv preprint arXiv:2310.19513.
> > >
> > > [3] Zhou, X., Xue, D., Chen, R., Zheng, Z., Wang, L., & Gu, Q. (2024). Antigen-Specific Antibody Design via Direct Energy-based Preference Optimization. arXiv preprint arXiv:2403.16576.
> > >
> > > [4]  Ye, F., Zheng, Z., Xue, D., Shen, Y., Wang, L., Ma, Y., ... & Gu, Q. (2024). ProteinBench: A Holistic Evaluation of Protein Foundation Models. arXiv preprint arXiv:2409.06744.

---

> > > > ### Comment · Reviewer_vhAX · 2024-11-21
> > > >
> > > > Until now, what I can see is $\Delta\Delta G = \Delta G_{ref-seq} - \Delta G_{gen-seq}$. The gap between two parts is supposed to be higher so that the gen-seq is more stable, and why  $\Delta\Delta G$ is the smaller, the better?

---

> > > > > ### Author Response · Authors · 2024-11-21
> > > > > **Thanks to the reviewer for the comment**
> > > > >
> > > > > We speculate that this might be a bug in OpenReview system. Here is a screenshot of our updated content, which we have uploaded to an anonymous online drive (https://drive.google.com/file/d/15whlD1l7-pm7NTUNTVQgGzZokRQ6i4Qw/view). We kindly ask you to try switching to a different browser and reopening the page for this paper to check if the updates are visible. We would like to emphasize that lower $\Delta\Delta G$ values indicate better results.

---

> ### Author Response · Authors · 2024-11-20
> **Thank the reviewer for the valuable questions and suggestions. Answer to Weakness and Questions Part3.**
>
> Reference:
>
> [1] Luo, S., Su, Y., Peng, X., Wang, S., Peng, J., \& Ma, J. (2022). Antigen-specific antibody design and optimization with diffusion-based generative models for protein structures. Advances in Neural Information Processing Systems, 35, 9754-9767.
>
> [2] Zhu, T., Ren, M., \& Zhang, H. Antibody Design Using a Score-based Diffusion Model Guided by Evolutionary, Physical and Geometric Constraints. In Forty-first International Conference on Machine Learning.
>
> [3] Shanehsazzadeh, A., Bachas, S., McPartlon, M., Kasun, G., Sutton, J. M., Steiger, A. K., ... \& Meier, J. (2023). Unlocking de novo antibody design with generative artificial intelligence. bioRxiv, 2023-01.
>
> [4] Shanehsazzadeh, A., Alverio, J., Kasun, G., Levine, S., Khan, J. A., Chung, C., ... \& Bachas, S. (2023). In vitro validated antibody design against multiple therapeutic antigens using generative inverse folding. bioRxiv, 2023-12.
>
> [5] Frey, N. C., Berenberg, D., Zadorozhny, K., Kleinhenz, J., Lafrance-Vanasse, J., Hotzel, I., ... \& Saremi, S. Protein Discovery with Discrete Walk-Jump Sampling. In The Twelfth International Conference on Learning Representations.
>
> [6] Høie, M., Hummer, A., Olsen, T., Nielsen, M., \& Deane, C. (2023). AntiFold: Improved antibody structure design using inverse folding. In NeurIPS 2023 Generative AI and Biology (GenBio) Workshop.
>
> [7] Shanker, V. R., Bruun, T. U., Hie, B. L., \& Kim, P. S. (2024). Unsupervised evolution of protein and antibody complexes with a structure-informed language model. Science, 385(6704), 46-53.
>
> [8] Kong, X., Huang, W., \& Liu, Y. Conditional Antibody Design as 3D Equivariant Graph Translation. In The Eleventh International Conference on Learning Representations.
>
> [9] Kong, X., Huang, W., \& Liu, Y. (2023, July). End-to-End Full-Atom Antibody Design. In International Conference on Machine Learning (pp. 17409-17429). PMLR.
>
> [10] Zhou, X., Xue, D., Chen, R., Zheng, Z., Wang, L., \& Gu, Q. (2024). Antigen-Specific Antibody Design via Direct Energy-based Preference Optimization. arXiv preprint arXiv:2403.16576.
>
> [11] Dreyer, F. A., Cutting, D., Schneider, C., Kenlay, H., \& Deane, C. M. (2023). Inverse folding for antibody sequence design using deep learning. arXiv preprint arXiv:2310.19513.
>
> [12] Ye, F., Zheng, Z., Xue, D., Shen, Y., Wang, L., Ma, Y., ... \& Gu, Q. (2024). ProteinBench: A Holistic Evaluation of Protein Foundation Models. arXiv preprint arXiv:2409.06744.

---

### Meta-Review · Area_Chair_d6eZ · 2024-12-21

**Metareview:**

The paper introduces a retrieval-augmented diffusion model, RADAb, for antibody design using structural homologous motifs to guide the generative process. It claims state-of-the-art performance in antibody optimization tasks.

Strengths include innovative integration of retrieval mechanisms with diffusion models and potential for enhancing biomolecular design.

Weaknesses could be limited experimental validation.

The decision to accpet is based on the paper's innovative approach in antibody generation.

**Additional Comments On Reviewer Discussion:**

Reviewer vhAX concerned with lower confidence about that antibody binding affinity optimization does not result in such a huge decline, as shown in previous works [1,2,3,4].

 [1]Kong, X., Huang, W., & Liu, Y. (2023, July). End-to-End Full-Atom Antibody Design. In International Conference on Machine Learning (pp. 17409-17429). PMLR.

[2]Kong, X., Huang, W., and Liu, Y. Conditional antibody design as 3d equivariant graph translation. arXiv preprint arXiv:2208.06073, 2022.

[3]Luo, S., Su, Y., Peng, X., Wang, S., Peng, J., and Ma, J. Antigen-specific antibody design and optimization with diffusion-based generative models. bioRxiv, 2022.

[4]Tan C, Gao Z, Wu L, et al. Cross-gate mlp with protein complex invariant embedding is a one-shot antibody designer[C]//Proceedings of the AAAI Conference on Artificial Intelligence. 2024, 38(14): 15222-15230.

---

### Decision · Program_Chairs · 2025-01-22

Accept (Poster)